# Tunable Terahertz Metamaterial with Electromagnetically Induced Transparency Characteristic for Sensing Application

**DOI:** 10.3390/nano11092175

**Published:** 2021-08-25

**Authors:** Jitong Zhong, Xiaocan Xu, Yu-Sheng Lin

**Affiliations:** School of Electronics and Information Technology, Sun Yat-Sen University, Guangzhou 510006, China; zhongjt5@mail2.sysu.edu.cn (J.Z.); xuxc5@mail2.sysu.edu.cn (X.X.)

**Keywords:** electromechanically, tunability, metamaterials, multi-functionalities, terahertz, refraction index sensor

## Abstract

We present and demonstrate a MEMS-based tunable terahertz metamaterial (TTM) composed of inner triadius and outer electric split-ring resonator (eSRR) structures. With the aim to explore the electromagnetic responses of TTM device, different geometrical parameters are compared and discussed to optimize the suitable TTM design, including the length, radius, and height of TTM device. The height of triadius structure could be changed by using MEMS technique to perform active tunability. TTM shows the polarization-dependent and electromagnetic induced transparency (EIT) characteristics owing to the eSRR configuration. The electromagnetic responses of TTM exhibit tunable characteristics in resonance, polarization-dependent, and electromagnetically induced transparency (EIT). By properly tailoring the length and height of the inner triadius structure and the radius of the outer eSRR structure, the corresponding resonance tuning range reaches 0.32 THz. In addition to the above optical characteristics of TTM, we further investigate its potential application in a refraction index sensor. TTM is exposed on the surrounding ambient with different refraction indexes. The corresponding key sensing performances, such as figure of merit (FOM), sensitivity (S), and quality factor (Q-factor) values, are calculated and discussed, respectively. The calculated sensitivity of TTM is 0.379 THz/RIU, while the average values of Q-factor and FOM are 66.01 and 63.83, respectively. These characteristics indicate that the presented MEMS-based TTM device could be widely used in tunable filters, perfect absorbers, high-efficient environmental sensors, and optical switches applications for THz-wave optoelectronics.

## 1. Introduction

Metamaterials are regarded as artificial materials that are remain undiscovered in the natural environment [1,2,3]. Due to their extraordinary properties, metamaterials are becoming an emerging field in physics, chemical, engineering, and electrics subjects. In the recent years, there have been many investigations and reports in various potential applications of metamaterials, such as cloaking devices, high-sensitive environment sensors, perfect absorbers, security screening, tunable ultrahigh-speed filters, imaging devices, high-efficient light emitters, and non-destructive testing [3,4,5,6,7,8,9,10]. Metamaterials show many unique electromagnetic properties including field enhancement [11,12], negative refraction index [13], artificial magnetism [14], electromagnetically induced transparency (EIT) [15], and so on. By properly tailoring the geometric parameters, metamaterials are able to be easily operated in a wide spectrum range that includes terahertz (THz), infrared (IR), and visible light [16,17,18,19,20,21,22,23,24,25,26]. Among the whole electromagnetic spectra, THz wave is the transition spectrum that usually occupies the spectrum in the frequency range from 0.1 THz to 10 THz, which is between the IR and microwave wavelength. Since that metamaterial has great and ultra-sensitive electromagnetic response in the THz frequency range, THz metamaterial has become an emerging field during the recent years. One of the most used typical configurations of THz metamaterial is a split-ring resonator (SRR), which is commonly a ring with a split. It was theoretically proposed in 1999 for the first time and experimentally verified in 2000. Since then, many derivative designs were investigated and demonstrated based on SRR, such as the complementary SRR (cSRR) [27], V-shaped SRR [28], U-shaped SRR [29], electric SRR (eSRR) [30,31], and three-dimensional SRR (3D SRR) [32,33], etc. However, when the metamaterial structure is fabricated on the traditional solid substrates, the resonant frequencies of metamaterial are usually unable to be tuned, which means that these designs can only absorb or filter certain electromagnetic spectra in a passive manner. Aiming to improve the flexibility and to enhance the electromagnetic response of the THz metamaterial, many literature reports focusing on tuning mechanisms were reported, such as ferroelectric material [34,35], laser pumping [36,37], electrostatic force [38,39], thermal annealing [40,41], liquid crystal [42], semiconductor material [43], and so on. In addition, micro-/nano-electro-mechanical systems (MEMS/NEMS) technologies can easily realize mechanical manipulation in micro-scale or nano-scale and, as a result, can hugely improve flexibility and enhance the electromagnetic response of MEMS-based metamaterial in the THz frequency range. There have been many reports on MEMS devices with different tuning mechanisms, such as the electrothermal actuator, electrostatic actuator, piezoelectric actuator, electromagnetic actuator [44,45,46,47], etc.

In this study, we propose and demonstrate a tunable terahertz metamaterial (TTM) based on the MEMS technology in the THz frequency range. This TTM structure is composed of an inner Au layer, which is called a triadius structure, and an outer Au layer, which is called a eSRR structure. The whole structure is fabricated on Si substrate. The inner triadius structures are connected to the MEMS-based electrothermal actuator (ETA). By driving different dc bias voltages, the height between the triadius and eSRR structures can be changed and, therefore, exhibit high flexibility. The geometrical dimensions of the proposed TTM are optimized, including the length and height of inner triadius structure and the radius of outer eSRR structure. The field strengths distributions in this study, including the electric (E) and magnetic (H) fields of the triadius structure, eSRR structure, and TTM structure, will be analyzed and discussed, respectively. In addition, in order to investigate the potential applications of TTM in the environmental sensing application, the key sensing performances of TTM, such as figure of merit (FOM), sensitivity (S), and quality factor (Q-factor), will be calculated and discussed, respectively. Additionally, while exposed on different-refraction-index (*n* value) environment, TTM shows highly linear sensitivity in terms of the *n* values. These unique electromagnetic characteristics indicate that the TTM structure can be widely used in THz-range application fields, such as filters, switches, and high-efficient environment sensors including gas sensors, biosensors, chemical sensors, etc.

## 2. Design and Method

The schematic drawings of MEMS-based TTM and TTM unit cell are shown in Figure 1a,b, respectively. TTM is composed of the triadius and eSRR structures. A 300 nm thick Au layer is used in TTM. The inner triadius structures are connected to MEMS-based electrothermal actuator (ETA), which could exhibit high flexibility by driving different dc bias voltages to bend downwards. Figure 1c shows the geometrical denotations of the TTM unit cell, including metal length (*L*), radius of eSRR (*R*), and height between inner triadius and outer eSRR structures (*h*). The metal linewidth and gap of the inner triadius and outer eSRR structures are 5 µm. Figure 1d plots the relationship of driving voltages and displacements of MEMS-based TTM. The inserted images of Figure 1d are the geometrical dimensions of ETA. Due to the different thermal expansion coefficients between different materials, the cantilevers would be upward-bending after the release process in fabrication. Therefore, by driving different dc bias voltages on ETAs, the reconfiguration of MEMS-based TTM was proposed in order to compare and discuss different *h* values. The deformation in the free end of ETA is inversely related to applied voltage. In order to actuate the TTM unit cell for bending downwards, a driving voltage with a maximum value of 0.45 V would be induced on the TTM device. The inserted images include the inner triadius structure on ETA without and with a driving voltage of 0.45 V, respectively. The initial height is *h* = 2.46 μm when the MEMS-based TTM is released, while *h* value will be bent to 0 μm when the driving voltage increases to 0.45 V. It is clear that the proposed MEMS-based TTM could be actively actuated to tune the resonance by bending the cantilevers downwards.

The optical properties of the proposed TTM device are simulated by using Lumerical Solution’s finite difference time domain (FDTD) based simulations. Here, we define TE mode when the polarization angle equals to 0° and TM mode when the polarization angle equals to 90°. The propagation direction of incident light is set to be perpendicular to the *x-y* plane in the numerical simulations. Periodic boundary conditions are also adopted in the *x*-axis and *y*-axis directions and perfectly matched layer (PML) boundaries conditions are assumed in the *z*-axis direction. The transmission spectra (*T*) are calculated by monitor set on below of device. In these configurations, Si material serves as the substrate with the tailored Au layer atop. The permittivity values of Au and Si materials in the mid-IR wavelength range are calculated according to the Drude–Lorentz model [48,49].

## 3. Results and Discussion

The transmission spectra of the triadius structure in TE and TM modes with different *L* values are shown in Figure 2, respectively. This triadius structure shows polarization-dependent characteristics. For example, when *L* = 47.5 μm, the TE and TM resonances are at 0.58 THz. With *L* value decreasing by 5 μm steps from 47.5 μm to 32.5 μm, both TE and TM resonances are increased by 0.16 THz, (from 0.58 THz to 0.74 THz). As plotted in Figure 3, the field strength distributions (E-fields and H-fields) of the triadius structure possess a *L* value of 42.5 μm when monitored at 0.58 THz in TE and TM modes, respectively. The E-field strengths are focused on the end of the triadius structure, while the H-field strengths are concentrated along the contour of the triadius structure.

The transmission spectra with different *R* values of eSRR structure in TE and TM modes are shown in Figure 4a,b, respectively. In Figure 4a, eSRR shows the EIT characteristic at 0.60 THz with *R* = 45 μm. The EIT resonance is shifted to 0.68 THz and the *R* value decreased to 40 μm. The shifting range of EIT resonance is 0.08 THz. This EIT resonance gradually vanishes by continuously decreasing *R* value to 30 μm. By decreasing *R* value in 5 μm steps (gradually from 45 μm to 30 μm), the resonances are blue-shifted with a shifting range of 0.40 THz (from 0.58 THz with *R* = 45 μm to 0.98 THz with *R* = 30 μm). On the other hand, in the TM mode, by decreasing *R* value in 5 μm steps (gradually from 45 μm to 30 μm), the resonances are modified with a modifying range of 0.32 THz (from 0.67 THz with *R* = 45 μm to 0.99 THz with *R* = 30 μm). According to the results mentioned above, eSRR exhibits polarization-dependent characteristics. Figure 5 plots the field strengths distributions (E- and H-fields) of eSRR with a *R* value of 40 μm in TE and TM modes, respectively. In TE mode, the monitors of field distributions are set as 0.67 THz, 0.68 THz, and 0.70 TH, while it is set as 0.74 THz in TM mode. It can be observed in Figure 5a–c that the E-field strengths are focused on the ends of arc-shape of eSRR in order to generate the electric quadrupole, six-polar, and dipolar modes at 0.67 THz, 0.68 THz, and 0.70 THz, respectively. Meanwhile, the E-field strengths of TM resonance are focused on the on the ends of the arc-shape of eSRR, which is the electric quadrupole mode as shown in Figure 5d. The corresponding H-field strengths of eSRR in TE and TM modes are shown in Figure 5e–h for the electric quadrupole, six-polar, and dipolar modes at 0.67 THz, 0.68 THz, and 0.70 THz for TE resonance and electric quadrupole mode at 0.74 THz for TM resonance, respectively.

The transmission spectra of TTM with different *R* values in TE and TM modes with a constant *L* value of 47.5 μm are shown in Figure 6a,b, respectively. The resonances are superimposed from the resonances of the triadius and eSRR structures. For example, when *R* = 45 μm, the second TE resonance is at 0.66 THz while the second TM resonance is at 0.70 THz, respectively. By decreasing the *R* value in 5 μm steps (from 45 μm to 30 μm), the second TE resonance is increased to 0.32 THz (from 0.66 THz with *R* = 45 μm to 0.98 THz with *R* = 30 μm), while the second TM resonance is increased to 0.30 THz (from 0.70 THz for *R* = 45 μm to 1.00 THz for *R* = 30 μm). Meanwhile, since the *L* value is kept as constant at 47.5 μm, the first TE and TM resonances influenced by the triadius structure almost remain unchanged. TTM shows the EIT characteristic at 0.60 THz with *R* = 45 μm in TE mode as shown in Figure 6a. The EIT resonance is shifted to 0.67 THz with the *R* value decreased to 40 μm. The shifting range of EIT resonance is 0.07 THz. This EIT resonance gradually vanishes by continuously decreasing the *R* value to 30 μm. The field strengths distributions (E-fields and H-fields) of TTM with *L* value of 47.5 μm and *R* value of 40 μm in TE and TM modes are plotted in Figure 7, respectively. In TE mode, the monitors of field distributions are set at 0.53 THz, 0.67 THz, and 0.73 THz, while in TM mode they are set at 0.54 THz and 0.78 THz. As observed in Figure 7a–c and Figure 7g–h, the E-field strengths are focused on the end of the triadius structure as well as the ends of the arc-shape of eSRR for both TE and TM resonances, respectively. Meanwhile, plotted in Figure 7d–f and Figure 7i–j are the corresponding H-field distributions of TTM in TE and TM modes (0.53 THz, 0.67 THz, and 0.73 THz for TE resonances; 0.54 THz and 0.78 THz for TM resonances), respectively.

The transmission spectra of TTM structure with different *h* values in TE and TM modes with a constant *L* value of 47.5 μm and a constant *R* value of 40 μm are plotted in Figure 8. The resonances are superimposed from the resonances of the triadius and eSRR structures. For example, when *h* = 0 μm, the first TE resonance is at 0.49 THz while the first TM resonance is at 0.51 THz, respectively. By increasing the *h* value from 0 μm to 3 μm, the first TE resonance is increased to 0.09 THz (from 0.49 THz for *h* = 0 μm to 0.58 THz for *h* = 3 μm), while the first TM resonance is increased to 0.07 THz (from 0.51 THz for *h* = 0 μm to 0.58 THz for *h* = 3 μm). Meanwhile, since the height of the eSRR structure is kept constant at 0 μm and the *R* value is kept constant at 40 μm, the second TE and TM resonances influenced by eSRR structure almost remain unchanged. Particularly, in TE mode, TTM with *R* = 40 μm shows the EIT characteristic at 0.67 THz. The field strengths distributions (E-fields and H-fields) of TTM with the *L* value of 47.5 μm, *R* value of 40 μm, and *h* value of 1 μm in TE and TM modes are plotted in Figure 7, respectively. In TE mode, the monitors of field distributions are set at 0.54 THz, 0.67 THz, and 0.73 THz, while in TM mode they are set at 0.54 THz and 0.78 THz. In Figure 9a–c and Figure 9g–h, the E-field strengths are focused on the end of the triadius structure as well as the ends of the arc-shape of eSRR for TE and TM resonances, respectively. Meanwhile, the corresponding H-field distributions of TTM in TE and TM modes are plotted in Figure 9d–f and Figure 9i–j, respectively.

In order to further explore the potential applications of the proposed TTM device in environmental sensing, the key sensing performances of TTM, such as figure of merit (FOM), sensitivity (S), and quality factor (Q-factor), are investigated. In this study, TTM with constant geometrical parameters (*L* = 47.5 μm, *R* = 40 μm, and *h* = 0 μm) is exposed on the surrounding ambient with different refraction indexes (*n* values). Figure 10a,b shows the trends of sensitivities between TE and TM resonances and *n* values, respectively. They are quite linear. Here, we define the corresponding resonances in TE and TM modes and the sensitivities as *ω*_1_, *ω*_2_, *ω*_3_, and *S* = Δ*f*/Δ*n*, respectively. The Δ*f* is the shift of resonant frequency and Δ*n* is the change of *n* value. In TE mode, the calculated *S* at *ω*_1_, *ω*_2_, and *ω*_3_ are 0.138 THz/RIU, 0.21 THz/RIU, and 0.379 THz/RIU, respectively. It is obvious that the third resonance (*ω*_3_) is more sensitive to the *n* value than the others. In TM mode, the corresponding *S* values at three resonances are 0.15 THz/RIU, 0.223 THz/RIU, and 0.373 THz/RIU, respectively. Obviously, these results indicate that the third resonance is more sensitive to the *n* value as well. The definition of Q-factor is that *Q* = *f_r_*/FWHM and FOM are defined as FOM = (1 − *A_r_*) × *Q* [50], where *f_r_* is the frequency of resonance and *A_r_* is the corresponding transmission amplitudes, respectively. The calculated Q-factors and FOMs values at different TE and TM resonances are plotted in Figure 10c,d, respectively. Table 1 is a summary table of the corresponding Q-factors and FOMs values. Let us take the third TE resonance (TE: *ω*_3_, green line) as an example, then the maximum, minimum, and average values of the calculated Q-factors are 72.47, 59.91, and 66.01, respectively, while the corresponding calculated FOMs values are 71.33, 56.49, and 63.83, respectively. The sensing performances of this design are better than those reported in literature reports [9,15,51] as summarized in Table 2. Therefore, the proposed MEMS-based TTM device could be suitably used in environment sensing fields, such as gas sensing, bio-sensing, and chemical sensing, etc. 

## 4. Conclusions

In conclusion, a reshaping TTM structure is presented and it is composed of triadius and eSRR structures. By tailoring the geometrical parameters of TTM, such as the length (*L* value) and height (*h* value) of the inner triadius structure and the radius (*R* value) of the outer eSRR structure, the corresponding electromagnetic behavior exhibits polarization-dependent, tunable bandwidth, electro-magnetically induced transparency (EIT), and large resonance-tuning-range characteristics. The variation of the *L* value causes the resonance blue-shift with a frequency range of 0.16 THz. While the variation of the *R* value shows that the transmission bandwidths could be modified to possess EIT characteristics. The variation of the *h* value shows that the resonance could be tuned 0.09 THz. In addition, by changing the surrounding refraction index (*n* value), MEMS-based TTM shows ultrahigh sensitivity to the surrounding environment. In TE mode, the calculated sensitivity value reaches 0.379 THz/RIU at most, the maximum Q-factor is 72.47, and the maximum FOM is 71.33. In TM mode, the calculated sensitivity value reaches 0.373 THz/RIU, the maximum Q-factor is 73.27, and the maximum FOM is 62.33. These results indicate that the presented MEMS-based TTM has great characteristics and great application potential for high-flexibility tunable filter, perfect absorber, imaging device, optical detecting, environment sensor, and switch applications in the THz frequency range.

## Figures and Tables

**Figure 1 nanomaterials-11-02175-f001:**
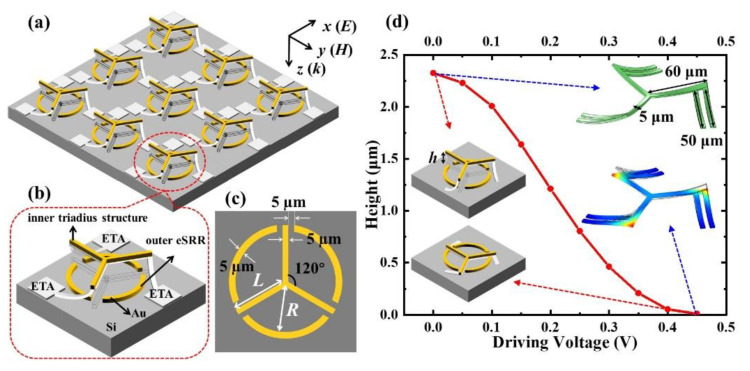
Schematic drawing of (**a**) MEMS-based TTM and (**b**) unit cell in detail. (**c**) Top view of TTM unit cell and the corresponding geometrical denotations. (**d**) Relationship of driving voltages and elevating heights (*h*) of TTM with ETA. Inserted images are the ETA simulations.

**Figure 2 nanomaterials-11-02175-f002:**
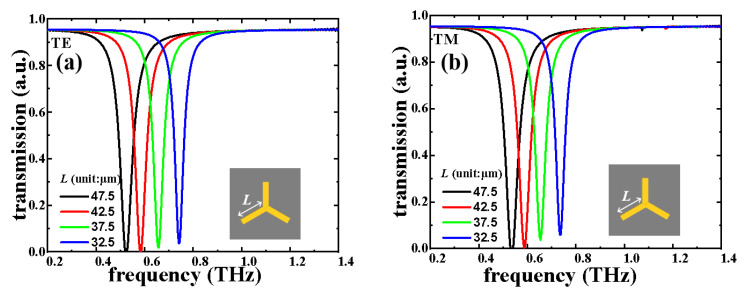
Electromagnetic responses of the triadius structure by changing the *L* parameter in (**a**) TE and (**b**) TM modes.

**Figure 3 nanomaterials-11-02175-f003:**
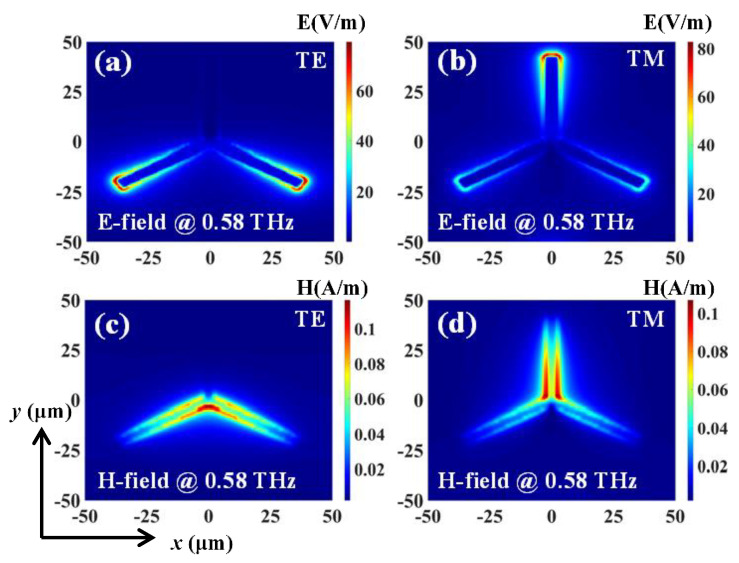
Field distributions of the triadius structure with *L* = 42.5 μm in TE and TM modes. (**a**) and (**b**) are E-field distributions. (**c**) and (**d**) are H-field distributions.

**Figure 4 nanomaterials-11-02175-f004:**
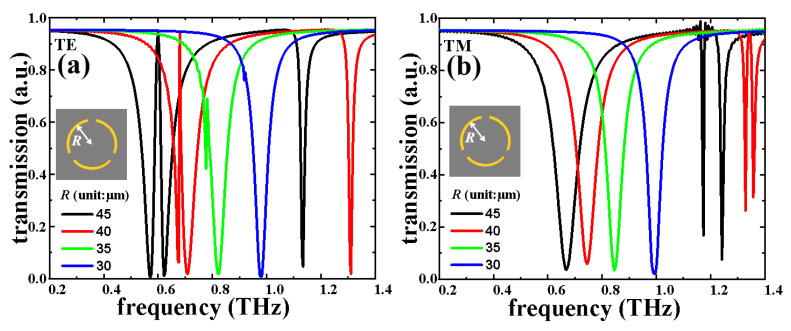
Electromagnetic responses of eSRR by changing the *R* parameter in (**a**) TE and (**b**) TM modes.

**Figure 5 nanomaterials-11-02175-f005:**
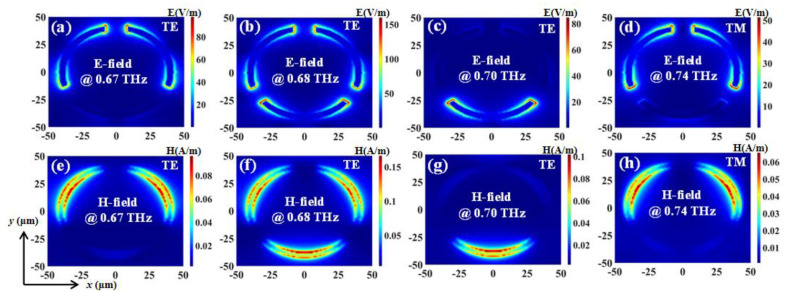
Field distributions of eSRR with *R* = 40 μm in TE and TM modes. (**a**–**d**) are E-field distributions. (**e**–**h**) are H-field distributions.

**Figure 6 nanomaterials-11-02175-f006:**
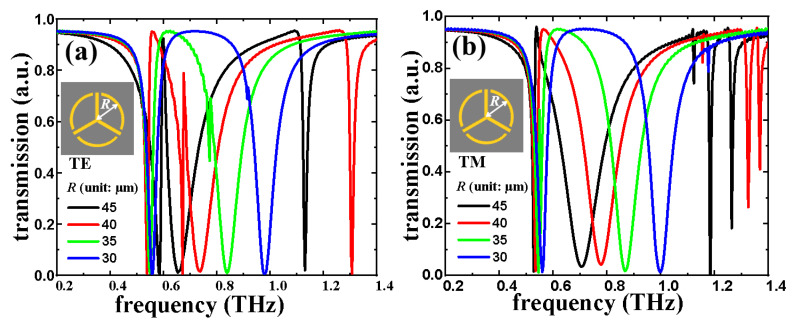
Electromagnetic responses of TTM by changing the *R* parameter in (**a**) TE and (**b**) TM modes under the condition of *L* = 47.5 μm.

**Figure 7 nanomaterials-11-02175-f007:**
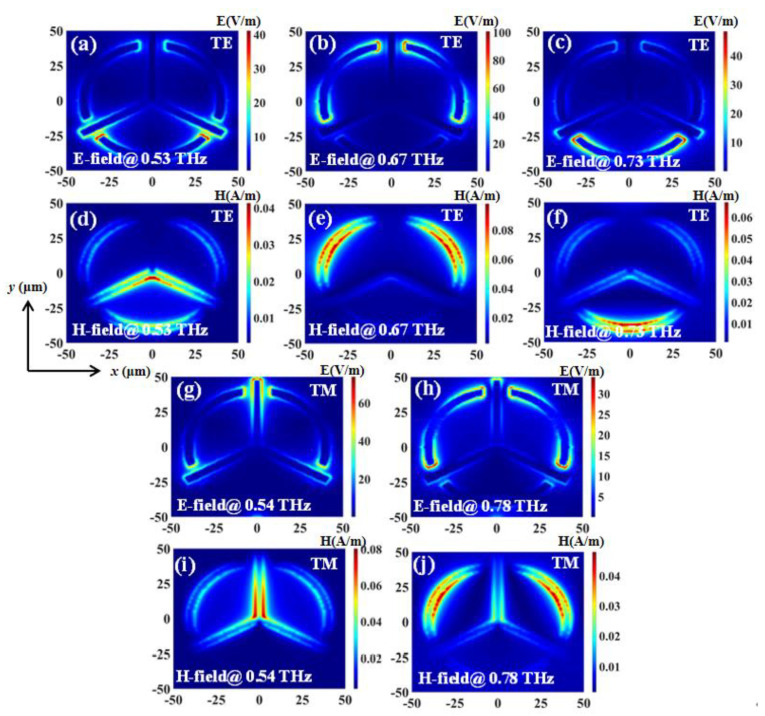
Field distributions of TTM with *L* = 47.5 μm and *R* = 40 μm in TE and TM modes. (**a**–**c**,**g**,**h**) are E-field distributions. (**d**–**f**,**i**,**j**) are H-field distributions.

**Figure 8 nanomaterials-11-02175-f008:**
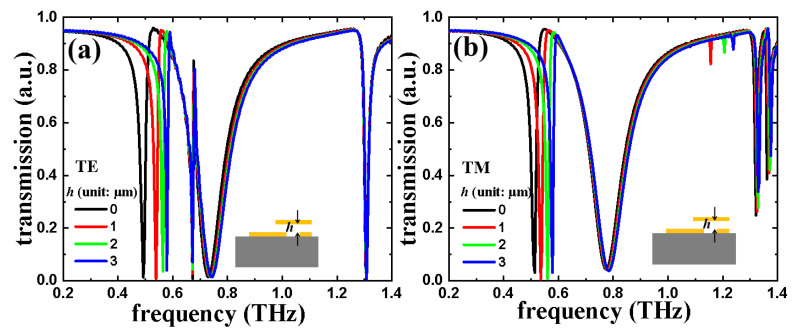
Electromagnetic responses of TTM by changing the *h* parameter in (**a**) TE and (**b**) TM modes.

**Figure 9 nanomaterials-11-02175-f009:**
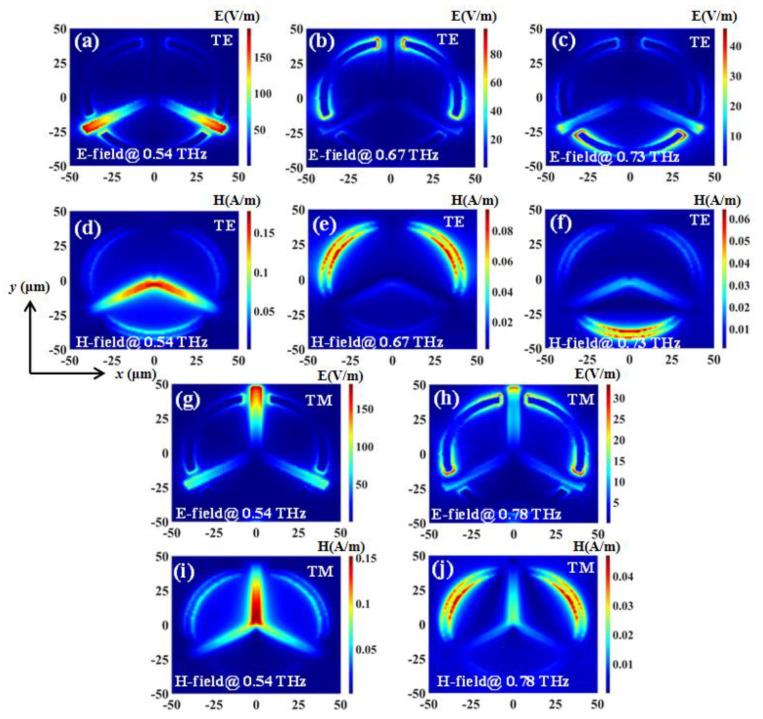
Field distributions of TTM with *h* = 1 μm in TE and TM modes. (**a**–**c**,**g**,**h**) are E-field distributions. (**d**–**f**,**i**,**j**) are H-field distributions.

**Figure 10 nanomaterials-11-02175-f010:**
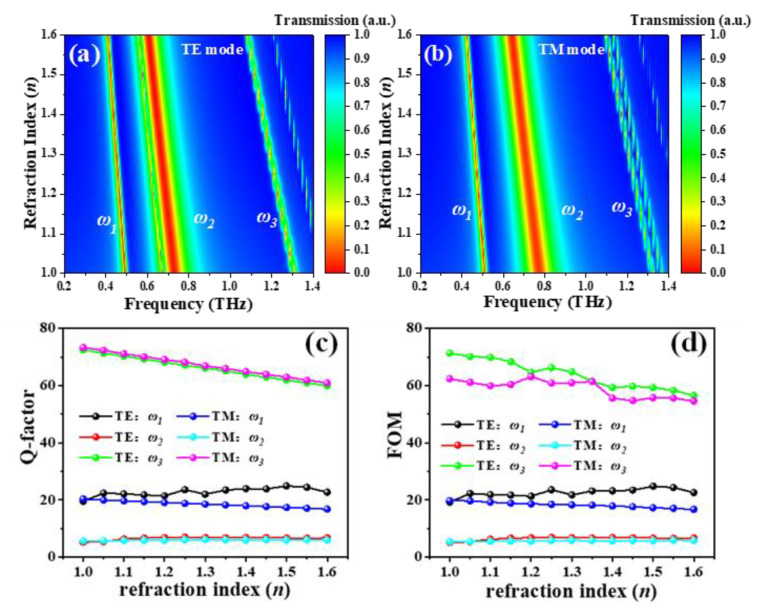
Electromagnetic responses of the TTM exposed on the surrounding ambient with different refraction index (*n*) in (**a**) TE and (**b**) TM modes, where *L*, *R*, and *h* parameters are kept as constants at 47.5 μm, 40 μm, and 0 μm, respectively. (**c**,**d**) are the Q-factors and FOMs of TTM in TE and TM modes, respectively.

**Table 1 nanomaterials-11-02175-t001:** Summaries of Q-factors and FOMs of TTM.

Resonance	Q-Factor			FOM		
	Max.	Min.	Ave.	Max.	Min.	Ave.
**TE: *ω*_1_**	24.89	19.49	22.76	24.78	19.22	22.54
**TE: *ω*_2_**	6.94	5.19	6.50	6.87	5.12	6.44
**TE: *ω*_3_**	72.47	59.91	66.01	71.33	56.49	63.83
**TM: *ω*_1_**	20.35	16.78	18.50	19.72	16.69	18.25
**TM: *ω*_2_**	6.11	5.64	5.90	5.84	5.40	5.66
**TM: *ω_3_***	73.27	60.91	62.10	62.33	54.51	58.92

**Table 2 nanomaterials-11-02175-t002:** The comparison of sensing performances in this study and literature reports.

	*ω* _1_	*ω* _2_	*ω* _3_	Reference
Sensitivity (S)	54.18 GHz/RIU	119.2 GHz/RIU	139.2 GHz/RIU	[15]
Sensitivity (S)	128 GHz/RIU	-	-	[51]
Sensitivity (S)	138 GHz/RIU	210 GHz/RIU	379 GHz/RIU	This study
Quality-factor (Max.)	11.6	-	-	[9]
Quality-factor (Max.)	24.89	6.94	72.47	This study
FOM (Max.)	2.30	-	-	[9]
FOM (Max.)	24.78	6.87	71.33	This study

## Data Availability

No new data were created or analyzed in this study. Data sharing is not applicable to this article.

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
