# Peer review of "Tunable Terahertz Metamaterial with Electromagnetically Induced Transparency Characteristic for Sensing Application"

_nanomaterials, 2021, doi:10.3390/nano11092175_

Round 1

Reviewer 1 Report

The paper describes a MEMS-based tunable metamaterial device for terahertz frequency range, composed of triadius-shaped and spilt-ring resonator gold inclusions on a silicon layer. The tunability is achieved by changing the distance between the triadius and the split ring by using an electrothermal actuator. The proposed design can be useful in sensing applications. The results of a detailed parametric study of the role of the various geometric parameters on the electromagnetic transmission through the structure are presented. It is unclear, how TE and TM polarization modes are defined for the wave which is incident normally to the surface. The direction of the E and H fields should be described by referring to the coordinate frame, along x or y axes. No separate electric or magnetic energy exists for time-varying fields, one should speak about electromagnetic energy or, even better, electromagnetic power. Figures 3, 5, 7 and 9 show no units for the field distributions. Poor English usage, significant editing is required.

Author Response

Please refer to the attached response letter and revised manuscript.

Reviewer 2 Report

The authors report here a computational study of metamaterial in terahertz region and its transparency characterization. They found that the structure shows metamaterial functionality and the frequency shift by the ambient refractive index change. The research content can be original and the results can contribute to the research field of the metamaterial. However, the reviewer found many concerns in the manuscript. After modification, the manuscript needs to be reviewed.

1) The research is a purely numerical simulation study, not an experimental one.  The reviewer does not find that the research is a numerical simulation-based study until he read “Results and Discussions” session. The authors should describe a main research method appropriately in “Introduction” and/or “Materials and Methods”.

2) There is no detail of the simulation method at all. The authors should describe the details;

  1. a) Method and software
  2. b) Polarization and orientation of the incident light. There is no definition of TE and TM.
  3. c) Literature of dielectric function of Au, SI, and an unknown material of ETA.
  4. d) Is ETA explicitly modeled in TTM simulation?
  5. e) Periodic or PML boundary condition? 1 looks periodic.

3) The current “Materials and Methods” is not related to the research content. The authors should fully change the description.

4) In all figures of electromagnetic field distributions, which fields do the authors display (E and H) or (E2 and H2)?  The authors should display the variables and the unit.

5) Why does EIT characteristics appear in Fig.4? What is the relationship between the structure and EIT frequency?

6) The frequency regions of Fig.10 and Figs.2-9 are different. The authors focus on ω3 but the peak frequency was firstly found at the final paragraph. Figs.2-9 should cover the frequency region and the nature of the frequency should be discussed if the ω3 is picked up at the refractive index sensing. Which frequency band do the authors want to focus on?

7) The authors compared their refractive index sensitivity with the refs 45-48. However, their refractive index sensitivities were not shown in the manuscript. It is difficult for the readers to compare the literature values with the results here. The authors should write them.

8) Frequency region of ref.48 (ACS Nano 2014) is different greatly from the manuscript. The detection technique of the spectra in ref.48 is also different. Furthermore, ref.48 does not show the refractive index sensitivity. Therefore, ref.48 is not comparable. The authors should cite more appropriate literature and compare their results with the other structures by similar mechanisms (metamaterial) or different mechanisms (normal material such as surface plasmon).

9) There are some English or terminology errors. “literature” is an uncountable noun (p.2, L.58); “the E-field energy” (p.5), “H-field energies (p.5)” etc… should be E-field strength (or amplitude) or H-field strength (amplitude), not energy. Field (E or H) is not energy (typically poynting vector or a power density).

Author Response

(The authors gave the same response as above.)

Reviewer 3 Report

The authors report a numerical study about a MEMS reconfigurable THz filter to be used also as sensor devices.

The manuscript is well written and easy to be read.

In my opinion the manuscript can be accepted with major revision after the addressing of the following comments.

  • The author should consider the following papers regarding the tuning mechanism of THz filters

IEEE Photon. Technol. Lett. 28, 2459 (2016)

  1. Appl. Phys., vol. 96, no. 8, pp. 4072–4075, (2004)

Applied Physics Letters 110 (14), 141107 (2017)

Photonics 2020, 7(3), 48

  • In page 3, line 87-88 should be: “A 300 nm thick Au layer is used…..”
  • In page 3, line 94 the authors refer to inserted image of Fig1 d as dimension of ETA. However in Fig. 1b the ETA are far away from gold structure.

Can the author explain in a better way?

  • Why the triadius structure shows polarization independent characteristic? It not a C4 symmetry and I would expect to be sensitive to polarization as for the eSRR.
  • Could the author explain the origin of EIT and why it vanish decreasing R?
  • Why the authors choice L=47.5 um for the simulation of Fig6?
  • The quality of Fig.6 should be improved.
  • In the text at page 7, the Figure number it is wrong. It is Fig.8 instead of Fig.7.
  • Why it is obvious that the third resonance is more sensitive?
  • Please compare the sensitivity with literature
  • In line 239, please add the literature value.
  • It is not very clear how the proposed filter can be experimentally fabricated. How the authors will fabricate suspended triadius?

Author Response

(The authors gave the same response as above.)

Round 2

Reviewer 2 Report

The authors have well modified their manuscript, and the reviewer suggests it is almost suitable for Nanomaterials in present form after just a minor revision.

Please check the physical units. All units must be written in SI form. For example, not “volts” but “V” (Capital) in 2. Design and Method session.

Author Response

Thanks for reviewers' comments and suggestions. We have revised the voltage unit as "V". 

Reviewer 3 Report

The authors replied to all of the critical points suggested by reviewers. I would recomend to accept this manuscript for pubblication.

Author Response

Thanks for reviewers' comments and suggestions.